# Applying a Health Access Framework to Understand and Address Food Insecurity

**DOI:** 10.3390/healthcare10020380

**Published:** 2022-02-17

**Authors:** Nasser Sharareh, Andrea S. Wallace

**Affiliations:** 1Department of Population Health Sciences, University of Utah, Salt Lake City, UT 84108, USA; andrea.wallace@nurs.utah.edu; 2College of Nursing, University of Utah, Salt Lake City, UT 84108, USA

**Keywords:** food insecurity, access barriers, causal loop diagram, nonprofit organizations

## Abstract

The prevalence of food insecurity (FI) in United States households has fluctuated between 10% and 15% for the past two decades, well above the Healthy People 2030 goal. FI is associated with increased use of healthcare services and the prevalence of multiple health conditions. Our current efforts to address FI may be limited by measures that lack granularity, timeliness, and consideration of larger food access barriers (e.g., availability of food providers and lack of knowledge regarding where to obtain food). If the Healthy People 2030 goal of reducing FI to 6% is to be met, we need better and faster methods for monitoring and tracking FI in order to produce timely interventions. In this paper, we review key contributors of FI from an access barrier perspective, investigate the limitations of current FI measures, and explore how data from one nonprofit organization may enhance our understanding of FI and facilitate access to resources at the local level. We also propose a conceptual framework illustrating how nonprofit organizations may play an important role in understanding and addressing FI and its intertwined social needs, such as housing and healthcare problems.

## 1. Introduction

Food Insecurity (FI) is the reduction of food intake, disruption of eating patterns, and downgrading the quality and variability of a diet necessary for a healthy life [1]. FI is an enduring problem in the United States (U.S.), affecting between 10% and 15% of households annually (Figure 1) [2]. People living with FI are at higher risk of depression, violence, obesity, hypertension, diabetes, substance use, suicide, and lower medication/treatment adherence [3,4,5,6,7,8,9]. FI is also associated with higher healthcare costs and increased use of healthcare services [10,11]. Since FI-related interventions and practices have failed to meet the Healthy People 2020 goal [12], the 6% goal for FI is retained in Healthy People 2030 [13].

Primary attempts to address FI in the U.S. are the U.S. Department of Agriculture (USDA) nutrition assistance programs, which aim to increase access to healthy food for low-income residents [14]. The largest nutrition assistance program in the U.S. is the USDA’s Supplemental Nutrition Assistance Program (SNAP) [15], covering 42.2 million individuals and costing $68 billion in 2017 [16]. Despite the USDA’s goal, many Americans still live in FI. This could be because federal nutrition assistance programs (1) leave food-insecure households with only slightly higher incomes than the cutoffs ineligible to participate [17]; (2) cannot logistically reach all low-income, food-insecure households [17]; and (3) do not always provide adequate benefits [18,19,20], specifically in times of economic uncertainty, such as during the coronavirus disease 2019 (COVID-19) pandemic [18]. These limitations compromise the effectiveness of federal nutrition assistance programs and emphasize the need for addressing food access barriers beyond offering financial assistance.

### 1.1. Food Insecurity through the Lens of Access Barriers

Having access to resources means that, when people are in need of a specific service, they be able to fully benefit from that service [21]. Over time, health services researchers have shed light on different barriers and indicators of service access [22]. While there are many models, the more common and enduring are Andersen’s behavioral model of health services use [23], Penchansky’s access model based on patient satisfaction surveys [24], and more recently, taxonomies of barriers of access to resources [21]. Using these models, barriers to accessing services can be categorized as predisposing characteristics (e.g., age, gender, education, occupation, ethnicity, genetics, psychological characteristics, and past illnesses; and attitudes, values, and knowledge about health and community resources), and community and personal enabling resources such as income and policies [23,25]. Society, institutions (e.g., schools and government), and individuals are responsible for these barriers [21]. Examples of societal barriers include inequalities posed by the society on the socioeconomic status of people, healthcare, housing, stereotyping, and general lack of understanding about the need for specific services; examples of institutional barriers include poor communication, outdated policies, administrative complexity, and physical and geographic barriers; and examples of personal barriers include stigmas, limited income, cultural beliefs, and family circumstances. To understand FI from an access barriers perspective, we categorized these examples of mutable FI-related access barriers—which are recognized in related literature [3,26,27,28,29,30,31,32,33]—into predisposing and enabling access barriers posed by society, institutions, and individuals in Table 1.

Access barriers are clearly interrelated. Economic instability, also independently associated with health outcomes, forces people to prioritize other issues such as housing and health over food needs [34,35]. Geographic and eligibility barriers to procuring community food resources (e.g., emergency food providers, healthy food options from supermarkets, federal nutrition assistance programs, etc.) are caused by transportation barriers [34], living far from a supermarket or food providers [36], and strict eligibility criteria and complicated application process of federal nutrition assistance programs [37,38]. Informational barriers could be personal such as lack of understanding about community resources (e.g., community food resources, primary care providers, and emergency housing assistance) and not knowing about where to get help, and could be societal when there is a lack of understanding about the required need for specific resources in a community. The stigma associated with participation in federal programs due to an aversion to receiving assistance or negative reactions and disapproval from peers can also impact access to federal resources [37,39].

### 1.2. Limitations of Food Insecurity Measures

FI-related measures and datasets are all limited by a lack of comprehensive consideration of the full range of access barriers, which are presented in Table 1; lack of granularity at the local level; and inadequate timeliness to be responsive to economic and public health emergencies. Efforts to understand FI at a population level have focused primarily on household surveys and the availability of food resources. While many different approaches [40] and surveys [41] exist, the official U.S. prevalence rate of FI is measured by the Current Population Survey (CPS) food security survey module [1,15,42]. The USDA measures FI through an annual supplement to the monthly CPS. However, CPS provides reliable data only at the state and national levels, and reliable estimates of FI from CPS at smaller geographic levels—where services are delivered to individuals—are not available [15,43]. Moreover, the CPS food security supplement focuses on the ability of individuals or households to afford/buy food [15,42]. and does not assess barriers to accessing federal or community resources such as lack of food providers, transportation, and knowledge regarding where or how to obtain food. In addition, since CPS is an annual survey, it cannot provide real-time data. Limited focus and lack of granularity and timeliness of CPS limit its ability to direct timely public health interventions that are delivered at the local level.

In contrast to CPS, Feeding America’s annual Map the Meal Gap study estimates FI rates at the county level vs. state level using seven different census variables including poverty rate (excluding undergraduate students to prevent reporting artificially high rates in college towns), unemployment rate, homeownership rate, median income, Hispanic percentage, African American percentage, and disability prevalence [33]. They use these indicators to measure the corresponding coefficients at the state level. Then, the same indicators at the county level and the state’s coefficients are used to estimate county-level FI. However, like CPS, Map the Meal Gap does not include the full range of access barriers and does not provide real-time data that can be applied to individual communities. As a result, these data are also limited in its ability to accurately direct timely food-related policies at the local level.

A more recent approach proposed for understanding FI is through data collected during healthcare encounters. FI assessments have been added as part of an expanded vision of assessing social determinants of health by Accountable Care Organizations, and have been encouraged by the Centers for Medicare and Medicaid Services and the National Academy of Sciences [44,45,46,47]. However, it is unlikely that data collected through this mechanism will be representative of a community’s FI status as many of those who experience FI are also those experiencing barriers to accessing health services. Further, data suggest that the social stigma associated with FI and other social determinants may limit the ability to understand the problem during healthcare encounters, apply for federal programs, and seek information about available resources [48,49].

Another co-existing problem with FI is the accessibility or availability of services such as supermarkets (i.e., “food deserts”) [50,51]. The USDA identifies food deserts by highlighting low-income areas where people must travel at least one mile to reach a supermarket or grocery store [36]. While this approach is informative (i.e., food deserts are likely associated with higher FI rates), it does not consider how access barriers may continue to reinforce FI: small grocery stores may offer only expensive unhealthy fresh food options; and many other economic, cultural, community, and social service factors (e.g., hours of operation, public transportation, income) may partly determine whether food can be procured [52]. In other words, even if food resources are available, they may not be affordable, acceptable, or accessible to individuals and families.

Collectively, the limitations of existing datasets and approaches suggest that moving from describing to understanding and addressing FI requires real-time information that is more geographically granular, and considers a wider range of access barriers as shown in Table 1.

### 1.3. Nonprofit Organizations’ Role in Addressing Social Needs including Food Insecurity

Many nonprofit organizations aim to address FI and other social needs such as housing and healthcare in the U.S. through local interventions, and by addressing a wide range of access barriers such as transportation challenges and informational barriers (see Table 1). These nonprofit organizations produce reliable, real-time data at the local level by way of tracking requests for social needs including food needs. We propose that this “information-seeking” for help from nonprofit organizations may be a powerful tool in unveiling unmet food needs at the local level where SNAP and other federal program participation might suggest needs are met [33,53]. Moreover, these data could identify individuals who are struggling with other social needs such as housing and healthcare, and may be able to do so in real-time, providing the opportunity to initiate targeted, timely public health interventions.

Two examples of nonprofit organizations whose data may pave the way for a more effective understanding of FI and other social needs are United Way’s 211 system (UW 211) and Benefits Data Trust (BDT). UW 211 is a national network of free-of-charge referral services to low and no-cost community resources [32] for a range of social needs including food, housing, and health services. For each 211 food-related telephone calls, callers’ demographics, needs, and referrals to food resources and programs such as SNAP, food pantries, and soup kitchens are collected, providing a new source of data that capitalizes on other dimensions of FI. Food-related calls are among the top three needs among callers and could be used as an indicator of unmet community needs [54]. In 2019 alone, UW 211 made 1.5 million connections to help address FI [54]. While food-related information-seeking—captured by consumer calls to UW 211—could be interpreted as another indicator of inability to afford sufficient food, incorporating information-seeking data into FI analyses might highlight regions where households with food needs and higher median household income (i.e., more ability to buy food) have limited knowledge about food resources in their community, or have troubles enrolling in federal nutrition programs. Similar to UW 211, BDT finds people in need and enhances access to essential government programs for nutrition assistance, housing, and healthcare [55]. BDT establishes targeted outreach or application assistance programs [29,30,31], and also impacts policy to simplify the application process for these programs. Since 2005, BDT has secured more than $7 billion in benefits for individuals in need of food, housing, and healthcare [55].

## 2. Methods

To systematically understand and address FI, we need to learn the complexities within the FI system. FI is a complex system [56] and a dynamic problem that is constantly changing. As noted previously, FI is intertwined with other important social needs such as housing and healthcare, and is interconnected with individual-level factors such as income and transportation barriers. These relationships between variables create feedback loops that simultaneously reinforce and balance FI rates. Creating a Causal Loop Diagram (CLD) [57] has been proposed as one way to understand and incorporate the important roles different variables play in a complex system from a conceptual perspective [57]. CLDs provide a big picture perspective of a complex system by visually representing protagonists (i.e., key variables) of that system and illustrating the causality among them. These protagonists are shown in text boxes, and the causal relationships are shown by arrows. Based on the literature, we developed a CLD to illustrate the causalities among access barriers and the FI system, incorporating nonprofit organizations as actors addressing access barriers, and identifying important feedback loops that could potentially explain the dynamic changes in FI rates. These loops are hypothesized to be responsible for the growth and decline of the system’s behavior, in this case, FI rates.

## 3. Results

Figure 2 highlights the associations between access barriers and FI, and the roles of nonprofit organizations in improving FI and addressing other relevant social needs to FI. These links have been identified using literature and studies of nonprofit organizations that are addressing FI in the U.S. [3,26,27,28,29,30,31,32,33]. Positive links indicate that changes in one variable lead to the same changes in another. Negative arrows indicate that changes in one variable cause an effect in the opposite direction in another. Important feedback loops in this system have been identified: regardless of the number of negative and positive arrows in a loop, if the number of negative arrows in the loop is even, this represents reinforcing (“R”) impacts on the system’s behavior—FI rate; if the number is odd, the loop has a balancing (“B”) effect. For instance, R1 in Figure 2 represents the Health loop where both FI and health outcomes impact each other (i.e., as FI increases, negative health outcomes increase, and vice versa). Additionally, it has been well documented that a stronger family structure with social capital and support leads to lower rates of FI [26,27]. However, even one person with FI or a healthcare problem can impact the well-being of other household members (Family Structure loop (R2) in Figure 2) [28]. In addition, B1 represents the Federal Nutrition Assistance Programs loop. In this loop, the surging rate of FI increases efforts to reduce FI and raises funding support for nonprofit organizations trying to address FI. These nonprofit organizations provide application assistance to individuals in need [29,30,31,32], which will expand access to these programs. Through federal programs, people will gain access to a diet necessary for a healthy life, which eventually will reduce FI. Moreover, housing problems force people to choose housing over food, which increases FI rates; however, these nonprofit organizations can also connect people to emergency housing resources, assisting people to overcome their housing problems (B4) [58]. Other loops highlighted in Figure 2 are defined in Table 2. To navigate through each loop, start from “Food Insecurity Rates” and follow arrows based on the explanation of that loop in Table 2.

In Figure 2, red variables are those with an exogenous impact on our system. We propose that FI researchers and other stakeholders could explore the system from this “bigger picture” perspective, and include these variables endogenously so that they will be entailed in feedback loops. However, since these variables need change at the policy level and are also difficult to address in the short term, they were not considered in our conceptual model endogenously. While the scope of this paper is presenting a conceptual model based on the literature, our proposed model could be converted to a simulation using techniques such as system dynamics simulation modeling [57], to explore different scenarios for policy analyses.

## 4. Discussion

Previous conceptual and mathematical models have explored socioeconomic factors, availability of healthy and unhealthy foods, food policies, and access to automobiles as contributors to FI [56,59]. However, to date, these models have not considered data from organizations connecting people with food resources and the role these providers play in the FI system. Further, these models have not accounted for other social needs and competing demands impacting the decisions of people with FI and, thus, they represent only a small range of factors serving as food access barriers. In this paper, we looked at FI through the lens of access barriers; investigated the limitations of current measure of FI; conceptualized the associations between FI and other social needs, such as housing and healthcare; and explored how data from unique nonprofit organizations may be used to understand and address food access barriers and associated social needs in the U.S.

Previous measures of FI have analyzed economic instability as the main indicator of food access barriers. However, they lack consideration of a wide range of access barriers, as shown in Table 1. Limited availability of food providers and supermarkets, lack of knowledge about available food options and about where to procure food (i.e., informational barrier), and ineligibility for federal nutrition assistance programs have a direct impact on FI [53,60,61]. The neglect of these access barriers in our current approaches and the lack of timeliness and granularity in current FI measures may contribute to a lack of understanding about FI at local level, and thus indirectly to the enduring FI rates in the U.S. (see Figure 1). These limitations likely restrict the impact of current data in efforts to address FI at the local level, as FI is multifaceted and subject to access barriers at the community level. In addition, the COVID-19 pandemic has further shed light on the importance of access barriers as food-insecure households have experienced higher access challenges during the pandemic compared to food-secure households [62].

Information-seeking behavior, as captured by nonprofit organizations such as United Way’s 211 and Benefits Data Trust, tells of many additional food-related access barriers, including informational, transportation, and limited food resources. Most importantly, while nonprofit organizations play a critical, central role in the FI system (as shown in Figure 2), their impact is undervalued and understudied. In a comprehensive analysis of United Way’s 211 system, it has been shown that, despite eligibility for federal nutrition assistance programs at the local level, significant food needs exist among community residents as food access barriers impede people from healthy food options [53]. Even in counties with lower rates of food-insecure individuals as reported by Feeding America [33], high rates of 211 calls for food resources could indicate unmet needs that would benefit from assessments of food access barriers or access to other community resources such as primary care providers and housing assistance programs. This discrepancy also suggests that our current FI measures are not able to distinguish existing unmet food needs and leave many people with access barriers such as transportation or informational barriers out of their estimations. More importantly, since information seeking from 211 is reported on a daily basis, it can reflect the status of food needs in real-time and address the lack of timeliness and granularity of current measures of FI. In other words, while current measures provide a big picture of FI at the county and state level, data from nonprofit organizations directly addressing clients’ food needs hold great potential for providing a detailed understanding of food needs at the local level where food needs can be addressed.

Our conceptual model illustrated in Figure 2 provides a big picture of the FI system in which factors impacting FI are also intertwined, therefore, addressing FI requires collaboration from many stakeholders. As illustrated in Figure 2, besides the food system, housing and healthcare are intertwined with FI and people struggling to pay for food are also experiencing health and housing issues [35,63,64]. Housing and healthcare issues create reinforcing feedback loops that trigger FI among households (see Figure 2 and Table 2) and emphasize the need for a systematic approach that attempts to resolve these social needs together. Getting access to food services through nonprofit organizations makes a huge difference to someone with FI on an individual level; however, how these organizations impact the FI system from a wider perspective through addressing other social needs associated with FI is unaccounted for and underrecognized. Moreover, because changes in 211 calls are immediately recorded and captured in real-time, they may also serve as an important indicator of emerging needs during times of public health and economic crises [53].

We believe reaching the Healthy People 2030 goal of reducing FI to 6% requires broadening our mental perception of FI in a way that can initiate interventions responsive to real-time public health needs and emergencies at the local level. This will not happen until we adopt new approaches in designing targeted, timely interventions. Nonprofit organizations that are enhancing services by holistically addressing multiple, intertwined social needs associated with FI; and tracking granular, community-level, real-time data, provides a key opportunity to both better understand and address FI in the U.S.

## 5. Conclusions

Current measures of FI lack consideration of a wide range of access barriers and real-time data. Access barriers range from predisposing factors such as informational barriers to enabling factors such as income and geographical barriers. We propose that studying FI through the lens of access barriers could provide more actionable, real-time insights for policymakers and stakeholders seeking to address FI. Besides demographic factors and access barriers, social needs also impact FI, as people need to choose between health, housing, and food needs. Therefore, supporting the ongoing service and data efforts of nonprofit organizations that are fighting against FI, housing issues, and healthcare problems at the local level will achieve the Healthy People 2030 goal of reducing FI to 6% and will improve public health. In addition, we illustrated the complexity within the FI system, which indicates the necessity of collaboration among various stakeholders to improve FI.

## 6. Limitations

Although the causal links illustrated in Figure 2 are not verified (but will be simulated and validated), they represent a reasonable picture of the FI system and how it is intertwined with access barriers and other social needs, giving important structure for efforts aiming to develop cost-effective solutions, empower nonprofit organizations, and reduce FI. While there are many nonprofit organizations of which our research team is not even aware, we presented United Way’s 211 system and Benefits Data Trust work. Although the awareness about these programs has increased over the years, as captured through increasing requests they receive every year from different communities [54], future work could investigate the validity and generalizability of their data. Finally, the association between existing unmet food needs, captured by calls to 211, with experiences of FI needs to be investigated.

## Figures and Tables

**Figure 1 healthcare-10-00380-f001:**
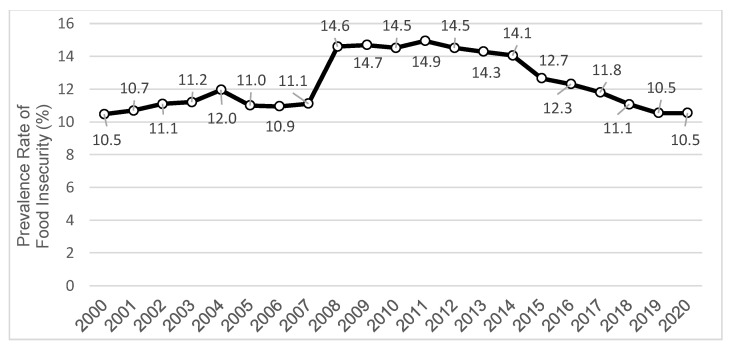
Trends in the prevalence rate of food insecurity (includes low and very low food insecurity) in U.S. Households—source: U.S. Department of Agriculture (USDA).

**Figure 2 healthcare-10-00380-f002:**
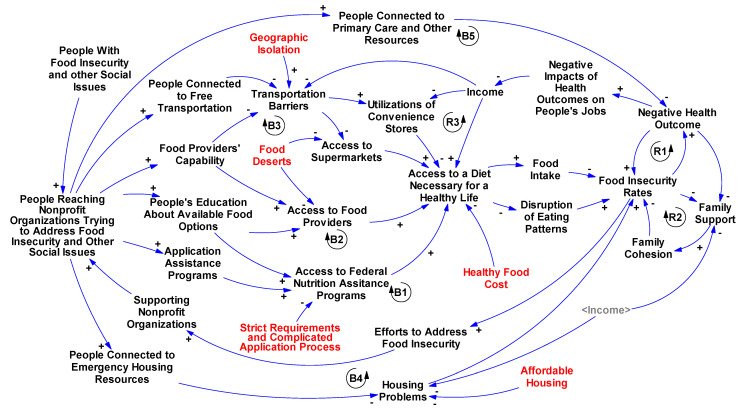
Causal Loop Diagram of access barriers and the role of nonprofit organizations in addressing social needs including food insecurity; red variables are exogenous variables that need policy changes. For illustration purposes, not all links are shown.

**Table 1 healthcare-10-00380-t001:** Mutable predisposing and enabling access barriers posed by society, institutions, and individuals that impact food insecurity.

	Predisposing Access Barriers	Enabling Access Barriers
**Societal-Level**	Inequality in education and occupation, Informational barriers (lack of understanding about the need for specific resources)	Inequality in income, healthcare, and housing
**Institutional-Level**	Attitudes towards food providers (e.g., food pantries, soup kitchens)	Food prices, Eligibility and application process of federal nutrition assistance programs, Geographic barriers (lack of access to supermarkets, and food providers, transportation barriers), Food policies, Poor communications about available food resources
**Individual-Level**	Stigma, Beliefs, Informational barriers (lack of information about resources), healthcare comorbidities	Economic instability (income), Social relationships

**Table 2 healthcare-10-00380-t002:** The list of highlighted reinforcing (R) and balancing (B) loops in Figure 2.

Loop	Name—Explanation
**R1**	Health loop: the causal link between FI and negative health outcomes such as obesity, depression, violence, etc. creates a reinforcing loop that increases FI and negative health outcomes
**R2**	Family Structure loop: FI creates stress, disorganizes households, and impacts family cohesion which reinforces FI.
**R3**	Unhealthy Food loop: Lack of income cause transportation barriers, which force people to acquire food from convenience stores with mostly unhealthy foods, and eventually lead to higher FI rates. Many other factors can also cause transportation barriers but we just illustrated the link between income, transportation, and FI in this loop. Additionally, economic instability impacts health and housing but the links are not shown.
**B1**	Federal Nutrition Assistance Programs loop: increasing efforts to address FI could include supporting nonprofit organizations to connect people to federal nutrition assistance programs or enhancing these programs’ capabilities in which both will provide critical benefits to low-income people to meet their food needs
**B2**	Food Providers loop: as the efforts to address FI increase, more nonprofit organizations assist food providers and will raise public awareness about these providers which will eventually raise access to food.
**B3**	Balancing transportation loop: some nonprofit organizations assist people to get access to free transportation, hencepeople could shop from supermarkets that are far from their neighborhood, which will decrease FI.
**B4**	Housing loop: nonprofit organizations can connect people to emergency housing resources, assisting people to overcome their housing problems, so that people will not need to choose between housing and food.
**B5**	Primary Care loop: addressing healthcare issues provide the chance to people to better perform their job duties, improve their economic stability, choose better food, andmeet their food needs.

## Data Availability

All the data supporting our paper is published herein.

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
