# Peer review of "Applying a Health Access Framework to Understand and Address Food Insecurity"

_healthcare, 2022, doi:10.3390/healthcare10020380_

Round 1

Reviewer 1 Report

Interesting research however, several points need to be taken into account before acceptance.

First of all, it is necessary to correct / explain / complete "Error! Reference source not found…” in lines 66, 84, 131, 135, 202, 241 and 261. The article seems incomplete.

Please add [%] in figure 1.

Lines 169-171: What is the difference between a Causal Loop Diagram (CLD) from 57 (Sterman J. Business dynamics: systems thinking and modeling for a complex world (Irwin/McGr). Boston, MA. 2000) and a qualitative CLD developed by you?

Lines 186-189: "Important feedback loops in this system have been identified: if the number of negative arrows in the loop is even, this represents reinforcing (‘R’)  impacts on the system’s behavior—FI Rates; if the number is odd, this has a balancing (‘B’) effect". In my opinion this must be explained. What if the number of negative arrows in the loop is even and at the same time higher (or lover) than the number of positive arrows in the loop? It doesn't matter at all?

Table 2, figure 2: In my opinion loops R1 and R2 are visible and correctly explained. Unfortunately, I can't see the others in the picture, for example B1 (I can’t see B1 “loop”). What’s more, in table 2, an explanation of loop B1: “Federal Nutrition Assistance Programs loop: receiving benefits from these programs provide a healthy diet to low-income people” and loop B1 in the figure 2 are inconsistent.

In the R3 loop in Table 2: "Income loop: economic instability impacts health, housing, transportation, and many other factors that eventually raise FI" –  transport is far from R3. Please explain.

Table 2: Missing R4.

Author Response

Thank you for the opportunity to revise and resubmit our manuscript ID#1545136, entitled " Applying a health access framework to understand and address food insecurity " for reconsideration by Healthcare.

We thank the editor and anonymous reviewers for their useful comments. We appreciate the opportunity to address the comments by submitting a revision. We have considered all comments. We believe the revisions address these comments and have greatly improved the paper.

Below, we have addressed each reviewer’s concerns by explaining in detail the responses (In bold) to their questions/concerns (numbered), and marking up the changes in the manuscript using the “Track Changes”.

a) Reviewer 1

  1. First of all, it is necessary to correct / explain / complete "Error! Reference source not found…” in lines 66, 84, 131, 135, 202, 241 and 261. The article seems incomplete

Thank you for pointing out this problem but, unfortunately, these issues do not occur on our end. However, we changed all the captions and references. We hope the changes fix the problems on your end. We have also asked the editorial team to double-check the manuscript to make sure the problem is fixed for reviewers.

  1. Please add [%] in figure 1

Figure 1 is updated accordingly. We added “%” in the vertical axis title.

  1. Lines 169-171: What is the difference between a Causal Loop Diagram (CLD) from 57 (Sterman J. Business dynamics: systems thinking and modeling for a complex world (Irwin/McGr). Boston, MA. 2000) and a qualitative CLD developed by you?

There is no difference. We added more details in this section so that readers without prior knowledge of CLD know that this diagram is a qualitative, conceptual model and not a quantitative, simulation model.

  1. Lines 186-189: "Important feedback loops in this system have been identified: if the number of negative arrows in the loop is even, this represents reinforcing (‘R’)  impacts on the system’s behavior—FI Rates; if the number is odd, this has a balancing (‘B’) effect". In my opinion this must be explained. What if the number of negative arrows in the loop is even and at the same time higher (or lover) than the number of positive arrows in the loop? It doesn't matter at all?

The number of signs will have no impact on the loop’s effect. For instance, one loop with 5 negative arrows and 2 positive arrows acts the same as a loop with 3 negative arrows and 6 positive arrows; both will have a balancing effect on the system’s behavior. We added a sentence in line 191 to address this concern.

  1. Table 2, figure 2: In my opinion loops R1 and R2 are visible and correctly explained. Unfortunately, I can't see the others in the picture, for example B1 (I can’t see B1 “loop”). What’s more, in table 2, an explanation of loop B1: “Federal Nutrition Assistance Programs loop: receiving benefits from these programs provide a healthy diet to low-income people” and loop B1 in the figure 2 are inconsistent.

For a complex diagram like Figure 2, it is inevitable to have multiple loops crossing each other. Also, for illustration purposes, not all links are shown. For instance, income also impacts housing but we have not added a link between them. The caption of Figure 2 is updated to point this fact.

Table 2 also assists readers to follow the loops. We added more information in Table 2 and line 207 to facilitate the reading of Figure 2.

Regarding B1, please see the descriptions in line 199.

  1. In the R3 loop in Table 2: "Income loop: economic instability impacts health, housing, transportation, and many other factors that eventually raise FI" –  transport is far from R3. Please explain.

Table 2 is revised to provide more clarity.

  1. Table 2: Missing R4.

Thanks for raising this problem. There is no R4 in our updated Figure4 now. Please see the revised Table 2.

Reviewer 2 Report

  • Please adapt the article to the journal's requirements.Please make the footnotes in square brackets.The year is shown in the bibliography.(temple attached).
  • Figure 1 - please add the numbers for each year in the chart (data labels)
  • Table 1. - please provide the source of the data development
  • Please correct “Error! Reference source not found.” Line 66, 84, 131, 202, 241, 261,
  • Method: Please describe this chapter in more detail.What static, econometric and mathematical methods were used to create the model.Where the primary data was taken from.
  • Results: Describe the conclusions drawn from Figure 2 in detail.
  • Results: Please insert map  with areas with high existing unmet food needs.
  • Discussion: Please enrich the discussion. Only 3 research papers were referenced in the discussion.
  • Conclusion: This entire chapter is for improvement. Please move the hypothesis to the introduction. The conclusions should include practical and theoretical conclusions resulting from the research results.

Author Response

Thank you for the opportunity to revise and resubmit our manuscript ID#1545136, entitled " Applying a health access framework to understand and address food insecurity " for reconsideration by Healthcare.

We thank the editor and anonymous reviewers for their useful comments. We appreciate the opportunity to address the comments by submitting a revision. We have considered all comments. We believe the revisions address these comments and have greatly improved the paper.

Below, we have addressed each reviewer’s concerns by explaining in detail the responses (In bold) to their questions/concerns (numbered), and marking up the changes in the manuscript using the “Track Changes”.

a) Reviewer 2

  1. Please adapt the article to the journal's requirements. Please make the footnotes in square brackets. The year is shown in the bibliography.(temple attached).

We adopted the MDPI Endnote style and updated the manuscript accordingly.

  1. Figure 1 - please add the numbers for each year in the chart (data labels)

Thanks for the suggestion. Figure 1 is updated accordingly.

  1. Table 1. - please provide the source of the data development

As mentioned in the paragraph above the table, all these parameters have been identified from literature [3,26-30] and have been categorized into mutable predisposing and enabling access barriers by the authors.

  1. Please correct “Error! Reference source not found.” Line 66, 84, 131, 202, 241, 261.

Thank you for pointing this problem but these issues do not occur on our end. However, we changed all the captions and references. We hope the changes fix the problems on your end. We have also asked the editorial team to double-check the manuscript.

  1. Method: Please describe this chapter in more detail. What static, econometric and mathematical methods were used to create the model. Where the primary data was taken from.

In line 171, we added more details about causal loop diagrams that they are conceptual, qualitative models and we included more information in that section to clarify the method. No mathematical equation is needed for developing such models. Also in line 188, we have mentioned the sources of those links “These links have been identified using literature and studies of nonprofit organizations that are addressing FI in the U.S”.

  1. Results: Describe the conclusions drawn from Figure 2 in detail.

The last paragraph of the Results section is now moved to the third paragraph of the Discussion section and more details are provided to enhance the conclusions.

  1. Results: Please insert map with areas with high existing unmet food needs.

This map has been already published and we do not have the authority to republish that. Please refer to “Sharareh, N.; Hess, R.; Wan, N.; Zick, C.D.; Wallace, A.S. Incorporation of Information-Seeking Behavior Into Food Insecurity Research. American Journal of Preventive Medicine 202”. We also deleted the last paragraph in the Method section to avoid confusion.

  1. Discussion: Please enrich the discussion. Only 3 research papers were referenced in the discussion.

We cited more papers that aligned with our research throughout the discussion. We also expanded the discussion section to address your concern #6.

  1. Conclusion: This entire chapter is for improvement. Please move the hypothesis to the introduction. The conclusions should include practical and theoretical conclusions resulting from the research results.

We revised the conclusion section as suggested.

Round 2

Reviewer 1 Report

The authors properly corrected the reviewer's suggestions and for this reason I am in favor of publishing the text.